# Economic Value Estimation of the Natural Heritage of the Tatra National Park

**DOI:** 10.3390/ijerph17093032

**Published:** 2020-04-27

**Authors:** Matus Kubak, Beata Gavurova, Klaudia Legutka

**Affiliations:** 1Department of Regional Sciences and Management, Faculty of Economics, Technical University in Košice, 040 01 Košice, Slovak Republic, matus.kubak@tuke.sk; 2Center for Applied Economic Research, Faculty of Management and Economics, Tomas Bata University in Zlín, 760 01 Zlín, Czech Republic; 3Faculty of Economics, Technical University in Košice, 040 01 Košice, Slovak Republic, legutka.klaudia@gmail.com

**Keywords:** contingent valuation method, willingness to pay, Tatra National Park, non-market goods, Slovakia

## Abstract

The aim of the study is to determine the economic value of the Tatra National Park. The willingness to pay approximation was used. Additionally, a questionnaire survey was applied in order to collect data. It contained a hypothetical event, and respondents expressed their willingness to pay an annual entry fee to the Tatra National Park in exchange for a guarantee of stopping the interference to its integrity. The total number of respondents was 921. The results show that the income level has a positive impact on respondents’ willingness to pay for entry to the Tatra National Park. With the increase of fee, the willingness to pay for entry to the Tatra National Park decreased by 2.2% for every additional price increase. The resulting value of the Tatra National Park, with the limits of the presented research mentioned in the paper, is approximately 17.5 million €.

## 1. Introduction

Value, in economics, represents the amount of a commodity that people may obtain in exchange for another commodity. Economic value, from the code of conduct point of view, is based on a criterion of one’s interest, i.e., a benefit that represents real economic value. This concept of self-interested actions of the economic subjects has a significant impact on nature and the environment. Historically, it has been shown that an absence of recognition of the importance of nature and the value that it contributes to society and economics led to underestimating nature and compromising services that are crucial to the well-being and prosperity of society.

The studies that focus on the evaluation of the social value of nature are able to inform social decision-making mechanisms, e.g., a state whose main role is to preserve and restore its natural resources. The primary aim of studies which focus on determining the value of nature is not to put ‘a price tag’ on the environment and its parts, but to convey an effect of a principal change in a group of ecosystem services, in case they would be exchanged for commodities with a certain value [1,2]. It is relatively simple to determine the value of ecosystem services that have clear ownership, such as wood or agricultural products. On the other hand, it is difficult to determine the value of ecosystem services that have a character of public goods and it is not possible to determine their ownership. 

Over time, the research of the valuation for non-marketed goods has developed into three branches. The first branch includes the methods of revealed preferences. These derive a value of non-market goods by observing a real (revealed) behavior at the market. They belong to indirect representatives of this method. The second branch consists of stated preference methods that consider the value of non-market goods by using the given behavior of individuals in a hypothetical situation. It is considered as a direct method of non-market goods’ valuation [3]. The third branch includes a method of economic efficiency that is frequently used for revenue and expense statements, values for physical, technical and/or economic parameters, which are required for effective modeling. 

Both methods, the willingness to pay (WTP) and the willingness to accept (WTA), are measuring such expenses and revenues. The contingent valuation method (CVM) is a method of estimating the economic value of non-market goods that are connected to the environment and nature. In most cases, it represents a type of public goods [4]. This valuation is performed via a survey questionnaire that obtains individuals’ preferences related to certain goods. Respondents express their preferences within WTP; acquisition or recovery, and/or preservation of the goods. Additionally, respondents may possibly be confronted with an alternative of the WTA status—either they would not be able to obtain a particular commodity or regularly derive the maximum benefit from it. A hypothetical market with particular goods is presented to individuals in order to determine these values. It results in the values of the WTP and the WTA that are both conditional on how a researcher describes this hypothetical market. Therefore, this process has been known as the CVM [5]. A consumer is asked to directly express the value of a researched commodity in a hypothetical context rather than to express this value in a real market behavior when using the CVM. Consequently, this method is considered as the most controversial one out of all methods that are used to state the value of non-market ecosystem services. Its application was the subject of this research. The main aim of this research was a quantification by using the CVM evaluation technique in order to create a potential for the ecosystem services’ support in the researched area—the National Park of High Tatras (hereafter TANAP), a national park in the north of Slovakia, a Biosphere Reserve of UNESCO. The analyses’ outputs are important for the creators of national and regional policies, especially in the processes of creating national and regional strategic development plans. Additionally, they provide valuable information for concepts of national and international benchmarking in this area.

## 2. Background of the Study

Nature is a valuable resource that every individual should protect because without nature, human life would not exist. The protection of nature makes sense especially nowadays, in an era of exponential economic growth and the trend of pursuing the highest profits. Mankind should realize the value of nature and reassess whether the social costs of decimating several hectares of forest can be balanced by the profit that an individual can achieve by selling harvested wood.

Excessive logging, deforestation and the construction of recreational and sports facilities in nature is an acute problem and environmental challenge in Slovakia. The majority of forests in Slovakia are state-owned. Management and use of state-owned forests are governed by forest management plans focusing primarily on forest production function which consists of the production of wood and other forest products. The intensity of forest use has increased over the last decade, with clearcutting being used extremely frequently. The abovementioned issues do not concern only ordinary forests, but also national parks. According to data from satellites [6], a similar percentage of forest coverage has decreased as in the neighboring countries of Slovakia, but forest loss is on average almost twice as large in the national parks of Slovakia. TANAP, which is the subject of the presented study, lost 30% of its forest from 2000–2016. The most important cause of forest loss in TANAP was the windstorm in 2004. In total, Slovakia lost 7% of forests between 2000–2016. Another issue is that the forests that disappear are old, natural, stable forests, which are complex ecosystems that are valuable in terms of biological conservation and biodiversity. The usual planting of a monocultural forest as a counterpoise of logging in a given area is far from compensating for this loss. TANAP, founded in 1949, is the oldest national park in Slovakia and together with the polish part of the Tatra National Park, High Tatras has been a UNESCO Biosphere Reserve since 1993. The national park area is 73,800 ha and its buffer zone covers another 30,703 ha. TANAP is visited yearly by 3.5 million visitors. The territory of the TANAP serves not only for its main mission, which is the protection of the extraordinary natural values of the territory but also for the needs of recreation, sports, treatment, and tourism. 

Despite the fact that TANAP is a national reserve and a UNESCO Biosphere Reserve, relatively intensive construction of hotels, sporting facilities, and facilities for tourists take place within the area. Various non-profit organisations, activists, home residents, and scientists opt for a slowdown of the construction process in the High Tatras. The existing ways of developing tourism centers and other facilities in TANAP lead to major biodiversity losses and disruption of natural processes. Habitats of European importance with a vast social value have already been irreversibly destroyed including in areas with fifth-level protection.

The question is whether the remaining part of the population (non-residents, the general public) would also desire the slowdown of construction activities and to what extent the non-residents appreciate TANAP. The answer might be outlined by the presented study, which focuses on the willingness to pay the hypothetical entry fee to the TANAP by non-residents in exchange for a guarantee of stopping the interference to its integrity.

## 3. Literature Review

Many foreign research studies dealt with the issues of the WTP and the CVM. Table 1 presents those studies that explicitly declare the use of the WTP and the CVM in the conditions of national parks, nature reserves, and protected sites.

As Table 1 shows, the target trajectory of mapped researches was identical. The authors especially used query techniques and a similar methodological apparatus that was related to the application of chosen methods. The differences were noticed in the size of the researched sample and in the way of its selection. It was determined by specificities of the researched groups and by a target of their research. The given research studies and their outputs were decisive in designing this research and in setting its interpretation platform.

Research indicates that there exist complexity and systematic context-dependency of the WTP determinants. Policy-makers and nature parks managers should be aware that visitors are often willing to pay much more than the present entry fee, which represents a significant potential for income increase to recover and maintain the parks. Even medium and low-income respondents are willing to pay for protecting significant environmental habitats. Results also indicate that women have a greater willingness to pay for the protection of nature and that moral norms and perceived control are the most significant predictors of visitors’ willingness to pay intentions. Many protected areas nowadays lack public financial help and a better understanding of the WTP may be the key to obtaining additional finance for nature protection.

## 4. Research Design

The collection of primary data was conducted from November 2016 to March 2017. Data collection was performed in person by personal polling of visitors to the Tatra National Park and in the form of internet polling on social networks focused on tourism, mountains, and the High Tatras itself. The survey was conducted from November 2016 to March 2017. There were 921 respondents’ answers collected. The questionnaire consisted of three parts: the first part aimed at collecting socio-demographic data about the respondents, the second part was designed to collect data about the reason and frequency of TANAP visits by respondents, and the third part of the questionnaire aimed at detecting respondents’ willingness to pay the hypothetical price per entrance to the TANAP.

Firstly, we propose a description of the first part of the questionnaire, thus the socio-demographic data. Out of 921 respondents in our dataset, 63% were females and 37% were males. The age of the respondents is the following: 61% of respondents were aged 19–35 years old, 33% were aged 36–62 years old, and 1.4% were aged over 62 years. The educational structure of respondents also varied. Most of the respondents reported as their highest attained level of education the secondary education with a general certificate of secondary education (37%). The second most frequent answer was a second university degree (32%), and the last group of respondents reported a first university degree (18%). Respondents from the Prešov region (21%) formed the largest group, the next group was formed by foreigners (15%), 13% of respondents came from the Košice region, and the smallest group of respondents came from Nitra and Trnava regions (6%). Income category was divided into four groups, with the first group including respondents with a monthly income to 405 € (28%), which represents the minimum wage in Slovakia in 2016. The second group of respondents’ monthly incomes ranged from 406 to 882 € (32% of respondents), with 882 € representing the average nominal wage in Slovakia in 2016. Another group of respondents ranged from 883 € to 1700 € (31% of respondents). As mentioned above, 883 € represented the average nominal wage in 2016 and 1700 was a median wage of employees within industries with the highest wages in Slovakia in 2016 (IT sector, finance and insurance, power engineering). The last group included respondents with incomes above 1700 € (9% of respondents).

The second part of the questionnaire collected data about the reason and frequency of TANAP visits by respondents. Multiple answers were possible. The most frequent reason for the visit was walking tourism due to which the TANAP is visited by almost 86% of respondents. The next reason was spending time with family (42% of respondents), 23% of respondents prefer skiing or snowboarding. Additionally, other activities included mountaineering, spa visits, ski mountaineering, cycling, and running. The second part of the questionnaire also examined the most frequently visited sites as well as the least visited ones, and those that are not visited at all. TANAP was divided into 10 sites that were described in more detail. and respondents were asked to mark the site that they have never visited or which they visit each time they go to TANAP on a scale from 0 to 4. The questionnaire provided the following results: the most frequently visited site is Štrbské Pleso by more than 49% of respondents who on a scale of visit frequency (0–4) marked 3 and 4. The second most visited site is Starý Smokovec, where 48% of respondents marked 3 and 4. The third most frequently visited site is Hrebienok and surrounding valleys and peaks, where almost 47% of respondents marked 3 and 4. Then, followed Tatranská Lomnica, surrounds of Tatranská Lomnica, Popradské pleso and the surrounding valleys and peaks, the Western Tatras, Ždiar and surroundings. The least visited sites are the valleys Tichá and Kôprová dolina, where more than 81% of respondents provided 0 and 1, while 54% of them have never visited these sites.

The third part of the questionnaire elicited respondents’ willingness to pay the hypothetical price per entrance to the TANAP. The respondents’ willingness to pay was elicited by the following question: *Imagine that the fee should be collected for an entrance to the Tatra National Park. Would you be willing to pay an annual amount (here the fee varied from 3–30 EUROS) for entering this area, in exchange for a guarantee that no interference with the integrity of the TANAP (further construction of hotels, cable cars or ski resorts) will take place and your payment will also cover repairs and maintenance of the Tatra National Park?* The main aim was to obtain approximately the same number of respondents’ answers for every hypothetical price for entrance to the Tatra National Park. This aim was accomplished and approximately 100 answers were obtained for each questionnaire type (price range). See Table 2.

Most of the respondents were willing to pay an entrance fee to the TANAP in exchange for a termination of any interference to its integrity. There were 65.2% of those surveyed who were willing to pay and 34.8% of respondents who would refuse to pay any entrance fee in the amount of 3–30 €. Those respondents who negatively answered on a given question were asked to give a reason. The following options were provided: (a) the Tatra National Park is not important for me, (b) I have a limited budget, (c) I do not think that I should pay for entering the national park, (d) I do not think that I am competent to give final judgment in this manner, (e) Other reason (provide some details). None of the respondents marked the first option, i.e., TANAP is not important for me. The most selected option was (c) that a respondent does not think they should pay an entrance fee to the national park. Almost 50% of those surveyed disagreed with paying any entrance fee. The most common response was that the responsibility of nature protection was by the state. Other responses that prevailed among respondents was that an individual should not pay anything for entering the national park as it belongs to everyone. Another group of respondents (18%) thought that they were not competent to give final judgment in this manner. Those respondents who had a limited budget formed 17% and the rest of the respondents (14%) provided other reasons. However, in the last option, there prevailed an answers expressing distrust that any financial means collected from entrance fees would be used for the given purposes and it would be spend in non targeted wey. The answer d) may be identified as a ‘strike’ answer. It is supposed, based on a theory, that a respondent does not provide a realistic approach to an expressed negative willingness to pay. Thus, such respondents were not taken into account in the model. The sample which was decreased by 10.6% of respondents was analyzed after some corrections and elimination of ‘strike’ and invalid answers. 

## 5. Materials and Methods

Cumulative distribution function 𝑊𝑇𝑃 is marked as 𝐺_𝑐_, and a related probability density function 𝑔_𝑐_ strongly depends on the type of questions used in the CVM questionnaire. In case of closed-ended questions, similarly, as the survey questionnaire of this research used, an individual is asked a question if s/he is willing to pay a certain amount 𝐴, and the probability that their WTP is equal or higher than a provided amount in the questionnaire is [21]:(1)$$\mathrm{Pr}\left(WTP\geq A\right)=1-G_{c}\left(A\right)$$

Many approaches are used to determine the WTP distribution. This study focuses on such an approach that is directly related to the main aim of its research. *Random Utility Model (RUM)* brings an aspect of randomness into the utility function. It is a basic model that is used for analyzing the dichotomous answers on questions in a questionnaire of contingent valuation. This model will provide an individual with his/her utility function (consequently, s/he will learn his/her WTP). However, an individual’s preferences are not observable for a researcher due to the nature of dichotomous questions. Therefore, the researcher treats the preferences as random variables, and consequently, an aspect of a mistake is directly involved in an indirect utility function. A probability that a respondent would answer ‘Yes’ may be specified as follows, while taking into account closed-ended questions of a contingent valuation [22]:(2)$$Pr\left(WTP=YES\right)=Pr\left\{WTP\left(q^{0},q^{1},p,\;y;\in \right)\geq A\right\}=Pr\left\{v\left(q^{1},p,y-A;\in \right)\geq v\left(q^{0},p,y;\in \right)\right\}=1-G_{c}\left(A\right)$$
where 𝑞^0^ and 𝑞^1^ are scalar expressions of goods or a service’s amount that are subject to valuation, while the initial state is (0) and the final state is (1), 𝐴 represents a monetary amount presented in a question. If $\mu WTP=E\left[WTP\left(q^{0},q^{1},p,\;y;\in \right)\right],$
$\sigma WTP^{2}=Var\left[WTP\left(q^{0},q^{1},p,\;y;\in \right)\right]$ and if 𝐺(∙) is a cumulative distribution function of a standardized variable $\omega =\left(WTP-\mu_{WTP}/\sigma_{WTP}\right)$; probability function may be rewritten to [21]:(3)$$Pr\left(WTP=yes\right)=\left\{1-G\left(A-\mu^{WTP}/\sigma WTP\right)\right\}=1-G\left(-\gamma +\delta A\right)=H\left(A\right)$$
where 𝛾 = 𝜇_𝑊𝑇𝑃_⁄ 𝜎_𝑊𝑇𝑃_ and 𝛿 = 1⁄ 𝜎_𝑊𝑇𝑃_. This expression, when a respondent’s answer on a closed-ended question is a function of a monetary amount, is in accordance with an economic model of maximizing behavior in those cases when it may be interpreted as a survival function of the WTP economic distribution. A probability model may be parametric or nonparametric if a relation between an offered price and a probability ‘Yes’ will not increase. Then a diagram that shows a probability function of all answers may be considered as a demand line after changing environmental goods or ecosystem services [22].

The most suitable model that was used in the analytical part of this study is logistic regression, which models a relationship between a nominal dependent variable 𝑌 and independent variables 𝑋.

The shape of a logistic regression model looks as follows: (4)$$\ln\left(\frac{Pr\left(WTP=yes\right)\;}{1-Pr\left(WTP=yes\right)\;}\right)=\beta_{0}+\beta_{1}x_{1}+\beta_{2}x_{2}+\ldots +\beta_{n}x_{n}$$
where ln(WTP = yes) describes a probability, variable 𝑌 that is dependent will be a ‘Yes’ value, while a probability of ‘No’ value is: Pr(WTP = no) = 1 − Pr(WTP = yes). Mostly the WTP = yes may be marked as a phenomenon that belongs to the 1st group, while the WTP = no is an opposite case, i.e., a phenomenon does not exist, it belongs to the 2nd group. The expression $\left(\frac{Pr\left(WTP=yes\right)\;}{1-Pr\left(WTP=yes\right)\;}\right)$ is marked as *odds* or a probability that a respondent will be willing to pay to a probability that s/he will not be willing to pay and its logarithm is marked as logit. 𝛽_0_ is a regression parameter, 𝛽_1_ … 𝛽_𝑛_ are unknown logistic regression coefficients, and their individual estimations are marked as 𝑏_𝑖_. 𝛽_0_ is a representation of a natural logarithm of a phenomenon probability. Consequently, mathematical editing of this expression will result in allocation probability to the 1st group.
(5)$$Pr\left(WTP=yes\right)=\frac{1}{1+e^{-\left(\beta 0+\beta 1x1+\beta 2x2+\cdots +\beta nxn\right)}}$$

The final predicted probability and estimation of logistic regression coefficients will be used for calculating the WTP expected value where [23]:(6)$$Mean\;WTP=\frac{1}{\beta 0}\times \ln\left(1+e^{\ln(1+e^{-\left(\beta 0+\beta 1x1+\beta 2x2+\cdots +\;\beta \mathrm{nxn}\right)}}\right)$$

The expected value of a willingness to pay for public goods is one of the other characteristics that express what amount of money a respondent will be willing to pay on the basis of the predicted probability of the logistic regression model. It is possible to determine the maximum expected willingness to pay (EWTP) and the minimum expected willingness to pay for public goods thanks to the predicted probability and its upper and lower bounds of reliability. The expression may be specified as: (7)$$EWTP=P_{i}\times Price_{i}$$
where Price_i_ refers to the price that was offered to each respondent *i*. Then, the 𝐸𝐴𝑊𝑇𝑃 (expected aggregate willingness to pay) for public goods may be calculated as a product of the expected willingness to pay that expresses a range of event’s impact. To be more specific, it may be a population that lives in a particular area, or territory, or the number of visitors as in this case. Mathematical relation may be written as follows [24]:(8)EAWTP=EWTP×number of visitors

### 5.1. Primary Data Processing—Analytical Platform Preparation

Table 2 illustrates a willingness to pay the entrance fee to the Tatra National Park depending on the amount of the entrance fee in the corrected sample.

Spearman’s correlation coefficient between an amount of entrance fee and the number of respondents who are willing to pay ρ = 0.681 (*p*-value = 0.03) and Spearman’s correlation coefficient between an amount of entrance fee and the number of respondents who are not willing to pay ρ = −0.734 (*p*-value = 0.016). The values of Spearman’s correlation coefficient show that there exists a relatively close relationship between increasing the amount of the entrance fee to the Tatra National Park and the number of respondents who are willing/not willing to pay for entering the Tatra National Park [25].

The number of respondents who are willing to pay for public goods should reflect the following based on the economic theory—the higher the price, the lower the number. The research results partially confirm this hypothesis, which is also evident in Figure 1.

The model of logistic regression includes an explanatory variable, which in this case is represented by an answer if a respondent is willing to pay a given price for entering TANAP in exchange for a guarantee of terminating any other interference to its integrity. Additionally, the research examined the influence of explanatory variables on respondents’ willingness to pay. Respondents had two options: yes/no that were modified into yes = 1, no = 0, while respondents had no knowledge of such a range of values provided in the questionnaires. It is supposed that households or respondents aim to maximize utility or income. This influences their decisions to pay for or refuse an offered price.

Variables, which were chosen as explanatory variables, should hypothetically influence the explanatory variable. Those explanatory variables that enter the model were selected based on socio-economic data of individual respondents by means of the survey questionnaire. Information on the purpose of visiting TANAP was not a part of the explanatory variables as more than 85% of respondents provided as their main reason being tourism. The price or the value of a willingness to pay is an explanatory variable that takes the following values 3–30 €. It is a randomly given price for entrance to TANAP that was approved or denied by respondents. It expresses a potential maximum value of a willingness to pay for annual access to the national park. Hypothetically, an increasing price should negatively influence a willingness of respondents to pay. Respondent’s sex is a dichotomous explanatory variable. The age of a respondent represents another explanatory variable. It is a categorical variable that divides a respondent’s age into three categories. It may be assumed that a younger respondent plans in a long-term period of time. Consequently, such a respondent attaches more importance to sustainability and nature protection. Therefore, a willingness to pay should be higher in the case of younger respondents. Respondents’ education was divided into seven groups in the survey questionnaire, while these groups became variables in the model and they were arranged in ascending order from primary education to higher education (third-level education). It is supposed that the level of education should positively influence a willingness to pay for public goods as it may be expected that a respondent with a higher level of education is more prudent in the environmental questions and also in the questions of sustainable development than a respondent with a lower level of education. The gross average monthly wage of a respondent is very important information about his welfare. Income was divided into four groups and it was arranged in ascending order from the lowest values to the highest in the model. The hypothesis is stated as follows, the amount of income will positively influence a willingness to pay for public goods and consequently, its increase will also increase a willingness to pay. In other words, poverty decreases the probability of a willingness to pay. The last explanatory variable of this model is a self-governing region, where respondents live. The fact that a respondent lives abroad was also considered. 

### 5.2. Model of Logistic Regression and its Parameters’ Estimation 

The model of logistic regression was used to estimate the influence of key parameters on the explanatory variable. The Hosmer-Lemeshow goodness of fit test (Table 3) was performed before the model’s estimation. The test is applied to samples larger than 400 observations. This condition was fulfilled as the sample consisted of 921 respondents. 

The test result is 𝑝-value > 0.05. It indicates the model’s suitability. Table 4 presents the output of the logistic regression model. This model estimated the odds of respondents from the individual categories who will be willing to pay an entrance fee to the national park, i.e., the odds that a variable ‘willingness’ will take the value ‘Yes’.

Respondent’s income, category of region and price of the entrance fee to TANAP represent statistically significant regression coefficients of the model according to *p*-value and the Wald test. In case of an income, a reference value was the first category that has values of gross monthly wage lower than 405 €. All income categories are statistically significant. A possibility that respondents whose income ranges from 405 €–882 € will be willing to pay for entrance to the national park, is 2.13 times higher than a possibility of respondents with an income lower than 405 €. The third category of income (883 €–1700 €) confirms the possibility that respondents who belong to this category will be willing to pay. It is 1.61 times higher than in the case of respondents from a reference category. In the case of the category where respondents earn the most, the possibility of a willingness to pay is 1.91 times higher than in the case of respondents from the first category. 

The hypothesis was partially confirmed, but it is not possible to declare that a higher income will result in a higher probability of a respondent who will be willing to pay an entrance fee to the national park. It seems that the lower-middle class is willing to pay an entrance fee to TANAP the most. These findings are in contradiction to the study of Togridou et al. [7] who state that income is not a significant predictor of a willingness to pay. On the other hand, other researches, such as Nuva et al. [9], Khan et al. [16], and Walpole et al. [11] confirm an income significance to a willingness to pay. 

The significance level 𝛼 = 0.05 reveals two significant categories from the ‘Region’ group. Reference category, in case of a region, where a respondent lives, has number 1—respondents who live in the Prešov region. The results show that a probability of respondents from the Košice region who are willing to pay an entrance fee to the national park is 1.47 times higher than a probability of respondents from the Prešov region. In the case of respondents from the Bratislava region, a probability of respondents who are willing to pay an entrance fee is 1.78 times higher than in the case of respondents from the Prešov region. Respondents who live abroad would have 1.36 times higher probability of a willingness to pay than respondents from the Prešov region. These differences among regions may result from a diversity of a living standard among the Slovak regions. The Prešov region is considered as the poorest region that may have a direct influence on a willingness to pay.

The price represents the most important variable whose coefficient was estimated by the model. It has a negative relation with regard to an explanatory variable. It means that when a price increases, a probability that a respondent would be willing to pay an entrance fee to the national park is lower by 2.2% for each additional increase of a price. Other variables were statistically not significant, i.e., significantly, they do not influence if a respondent is or is not willing to pay an entrance fee to the national park. Therefore, these variables will not be interpreted. Figure 2 illustrates the odds that were expressed by exposure of the predicted logit of the individual respondents. The mean was used to smooth a curve. The curve describes a situation when price increases, the respondent’s willingness to pay decreases. 

### 5.3. Expected Aggregation of Willingness to Pay

The value of an average willingness to pay was calculated based on the estimated coefficient of the logistic regression. The coefficients of explanatory variables with *p*-value < 0.1 were used to calculate this value. If the research took into consideration independent variables that are not statistically significant, it would significantly distort the final value. The final average value of a willingness to pay an entrance fee to the national park was determined in the amount of 15.92 €. Similarly, the given patterns were used to calculate the expected value of a willingness to pay, where the maximum and minimum expected value was determined. Figure 3 shows the results.

The mathematical expression of the expected willingness to pay an entrance fee to the national park provides a value in the amount of 10.90 €. This price will be used for the aggregate willingness to pay. The value will be calculated according to this pattern: 𝐸𝐴𝑊𝑇𝑃 = 𝐸𝑊𝑇𝑃 × number of visitors. At present, in Slovakia, there exists statistical capacity and performance of accommodation according to regions and districts [26], but there is an absence of data of the national park visitors. This research faced an issue that there only exists estimations of tourist numbers in TANAP. However, there does not exist any mechanism that would record numbers of visitors during a year. The TANAP administration performs once per year a census especially during summer, but more aggregate statistics are only estimated. In this research, the indirect method was used in order to calculate the number of visitors. The indirect estimation methods estimate the number of visitors based on observation of records collected for other purposes—guestbooks, parking data, ticket sales, etc. [27]. Historical press data where authors of the articles state an estimated number of domestic tourists per year in TANAP estimate approximately 1.1 million visitors annually [28].

The final aggregate value of the expected willingness to pay is €11.99 million [29]. The value will be €17.5 million when using an average willingness to pay that is expressed by means of coefficients. This result may not be considered as the aggregate value of TANAP, because a range of values is insufficient in terms of a willingness to pay (scale of hypothetical entrance fees to TANAP could have been wider in the questionnaire). The probability that a respondent would pay even the highest offered price is relatively high. Consequently, if a range of the offered amount of fees increases, the final value would be higher. These findings correspond with the results of the following research by Walpole et al. [11], and also with a willingness to pay even higher fees. An interesting comparison would be a distribution of questionnaires with the same questions, differentiated in a way of collecting the fees, to the respondents, and determining the final value. However, it is questionable if respondents would express a higher willingness to pay an entrance fee to TANAP in the case of it being a voluntary contribution or on the contrary, in the form of tax. These research questions are subject to further research.

## 6. The Economic Value of Natural Heritage, and Open Innovation in the 4th Industrial Revolution

OECD defines the eco-innovation as the implementation of new or upgraded products, services, processes, marketing methods, organization structures, and institutional set up which intentionally, or as the side effect, improve the environment [30]. The definition of innovation has a broader dimension, as it reflects the current needs of sustainable development and the so-called green economy.

The results of the presented study have a possible intersection with the areas of eco-innovation. The Slovak Republic takes part in the field of sustainable development policies and activities, including eco-innovation. The Slovak Republic also has a fair position in the Eco-Innovation Index among European countries. The innovative capacity of countries is currently an important geopolitical parameter and is examined through the innovative performance of the economy. Although there exists various tools to support the innovative performance of firms, public funding is considered to be most appropriate to strengthen research collaboration and networking of firms and public research entities. In addition to public resources, other options need to be sought to support ecosystem services and environmental sustainability.

Economies are constantly looking for suitable strategies not only to increase the competitiveness of companies but also entire industries. Economies must respond flexibly to global changes and changes in demand. It is innovations that, in the long run, make it possible to initiate new economic and technological cycles and achieve technological change. Only countries that have been able to identify and support viable innovations have secured the competitiveness and prosperity of its economies. Innovations also improve the development of regions, eliminate regional disparities, and improve the socio-economic position of companies. Countries that want to achieve the required macroeconomic and microeconomic goals must create suitable development conditions for the implementation of pro-innovative instruments while identifying perspective sectors. Innovation, entrepreneurship, cultural and creative industries are currently being assessed in a broader social, environmental, and societal context. Various globalization effects, as well as the processes of demographic aging that are putting pressure on the urgent issue of the sustainability of health and social systems, influence the abovementioned. Environmental aspects are an important part of the evaluation of socio-economic results from eco-innovation use. Environmental aspects have a direct link to changes in employment, exports and are also closely linked to eco-innovation. Eco-innovation can have a double positive effect on resource efficiency: it can increase the created economic value and at the same time reduce the pressure on the natural environment. An important element of the innovation process is social capital, examined in the context of social relations.

The presented study has some common topics with eco-innovation because it focuses on to the possibilities of exploring and evaluating economic dimensions of the use of natural resources and revealing several economic and non-economic potentials of natural resource development. An active endeavor on this topic will also support the development of regional innovation policies, as natural resources are usually bound to a geographically limited area. There is no one-size-fits-all innovation policy for the development of all regions. It is necessary to set up such support tools for given regions, which would take into account the specifics and developmental stages of the given region. Each type of region requires a specific approach, and the transition to higher levels of innovation can be stimulated and induced by appropriate supportive policies. 

The results of our study also point out the need for implementation of the new indicators, enabling the quantification of economic and non-economic benefits from the development of natural resources and the possibility of revealing new economic potentials in eco-innovation. These aspects are also directly related to the concept of creating the smart specialization strategies, preferred by the EU, whose main platform is the regional science and research assets, and the economic structure of the region. This will, even more, emphasize the importance of resources and knowledge in regional systems, as they are an essential part of innovation. Innovation will be gained through a combination of these resources and the ability to use and apply them. This will also support the further development of innovative approaches that can be the result of country innovation strategies. 

Nature as a natural asset should be protected by entire humankind. Humans cannot survive without nature. People should primarily recognize nature’s value and review the devastation of several square acres of forest due to, e.g., sale of logged timber, and consider if this action is worth the profit at the time of exponential economic growth and trend of monitoring the highest profit. This study did not focus on a calculation of nature’s value but on an examination of the value that respondents attribute to it in the researched natural habitat. Consequently, the research was realized on these bases. Its main aim was to allocate the potential of using the methods of a willingness to pay and a condition assessment to estimate the economic value of non-profit goods in TANAP.

However, in the Slovak conditions, there is an absence of such studies that apply these methods. Practice shows that the Slovak Republic belongs to the countries with the lowest rate of total expenses on environmental protection per GDP in the European Union [31]. This fact highlights the significance of the research study realization. The subject of this research was a willingness to pay an entrance fee to TANAP in exchange for a guarantee of terminating any other interference to its integrity. The condition assessment method based on the survey questionnaire was used. Respondents were introduced to a hypothetical situation when they were supposed to pay an entrance fee to the national park in the amount of 3 € to 30 €. The question in a form of dichotomous choice was asked, while the randomness of a sample was ensured. Almost 65% of respondents expressed their willingness to pay an entrance fee to TANAP. Subsequently, the model of binary regression was applied. The results show that statistically significant variables at the significance level 0.05 were the following variables: income, region a respondent lives in, the amount of entrance fee. The amount of income has a positive influence on respondents’ willingness to pay an entrance fee. If a variable describes a respondent’s origin, there was evident a positive relationship between respondents from the Bratislava region, the Košice region, and abroad. There was a higher rate of odds in respondents who were willing to pay an entrance fee to TANAP as opposed to those from the Prešov region. Finally, it was determined that the higher the entrance fee, the lower the willingness to pay an entrance fee to the national park. 

Average willingness to pay was calculated on the basis of estimated coefficients of the binary regression in the amount of 15.92 €. In case the mathematical calculation of expected willingness to pay was applied, the value was lower (10.90 €). The final value, that would be obtained after implementing the entrance fee to TANAP, is approximately €17.5 million or €11.99 million (in case of the second estimated price). The given value may be even higher if a range of offered prices is increased on more extreme values. As the questionnaire reveals, real implementation of an entrance fee to TANAP would bring many discrepancies. Generally, people do not believe that the fee would be used for a specific purpose. Thus, the use of entrance fee should be targeted and transparent to the visitor. This research state that the respondents would pay 15.92 € for entrance to TANAP, if it was guaranteed that no construction activities within this area would continue. The government bodies are responsible for respecting such guarantees.

## 7. Conclusions

The research results indicate that an application of methods that enabsles examination and evaluation of economic dimensions of natural resources’ use brings a valuable knowledge that may help to reveal possibilities of obtaining and effective use of potential resources for the renovation of ecosystem services in a given habitat. Examination of relations between a willingness to pay, income categories, and subjective perception of environmental wealth and utility from visits emphasized a research significance of visitors’ satisfaction and a reflection on their increasing expectations from visits in these habitats. Therefore, this information is inevitable in providing feedback on the process of effective use of environmental wealth and in increasing the number of visits to these habitats. Consequently, the management of natural habitats will actively start to manage the processes in the habitats, and expand facilities and services for visitors. Respondents emphasize a higher willingness to pay for future generations. It is an interesting fact that it is important to interconnect the education processes. As a consequence of these results, the environmental wealth value is differentially perceived in various educational categories. Thus, it is necessary to create and implement new concepts of environmental education in Slovakia that would increase awareness of the environmental health values in the country, their significance, the difficulty of protection and sustainability. Their positive effect will be evident in many economic and also non-economic dimensions. 

The results of this study put pressures on the creation and testing of suitable programs for entrance fees’ payment of a habitat based on income and geographical categories. This fact is justified by evident tendencies of public authorities to eliminate the rate of contributions to their maintenance and future development. The significant role of the national and regional policies that focus on the natural resources’ protection and environmental protection have come to people’s attention these days. The submitted study has two limitations. Firstly, the scale of the hypothetical entrance fees to TANAP could have been wider. Secondly, even if the submitted study deals with a relatively high number of respondents, it does not guarantee that the results would be possible to generalize for the general public. A random choice would be a guarantee. However, it is not possible to ensure it in these types of studies. The submitted study analyses the answers received. Consequently, a generalization of the results received would be a mistake in their interpretation.

## Figures and Tables

**Figure 1 ijerph-17-03032-f001:**
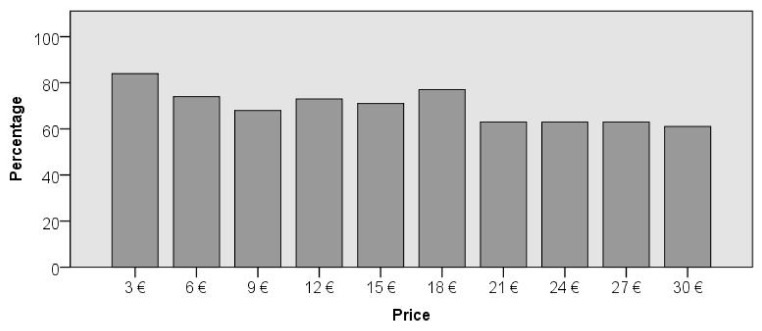
The percentage of respondents’ number who are willing to pay a given price. Source: own processing based on obtained data.

**Figure 2 ijerph-17-03032-f002:**
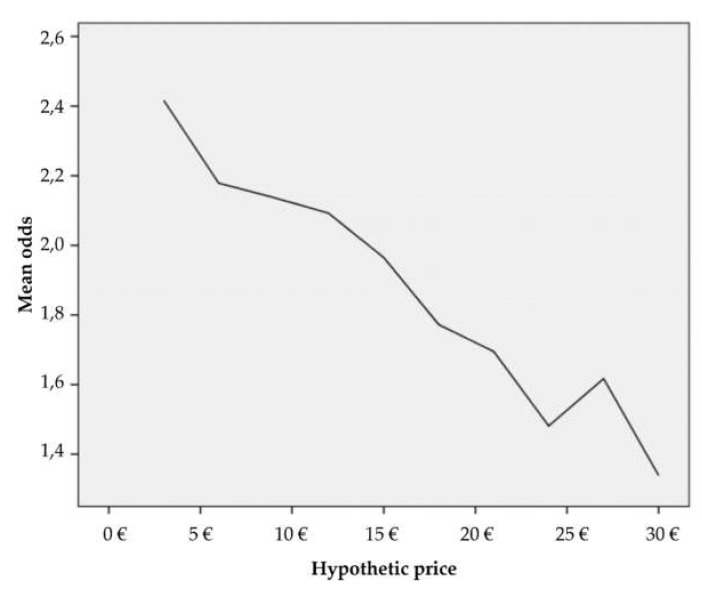
The odds’ graph with regard to the price. Source: own processing based on obtained data.

**Figure 3 ijerph-17-03032-f003:**
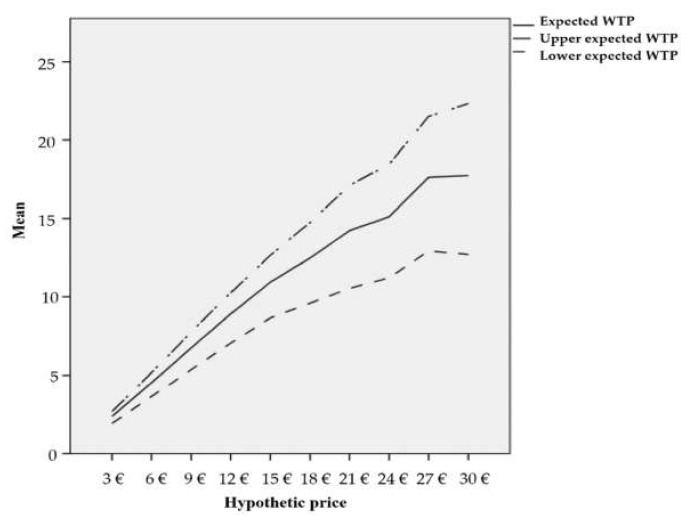
Maximum, medium, and minimum expected willingness to pay. Source: own processing based on obtained data.

**Table 1 ijerph-17-03032-t001:** Overview of foreign research studies of the WTP issue [explanatory notes: the WTP—Willingness to Pay, the CVM—Contingent Valuation Method, the OLS—Ordinary Least Squares].

Author/Year	Study Aim	Number	Methods	Final Findings
Togridou et al. [7]	Research of the impact of visitors’ profiles, information resources, environmental dispositions, and the WTP evaluation on Zakynthos visitors.	484	Questionnaire A question of ‘a principle of payment’ was used to find out respondents’ agreement with at least a partial payment. The method of open payment and regression analyses were also used.	The study refers to the complexity and context-dependency of the WTP determinants. Evaluation parameters of the visits were the most significant predictors. Incomes do not represent any significant predictor in the WTP answers, similarly as nationality. Park management suggests organizing some environmental education programs that could also represent feedback for authorized institutions. The authors also evaluate the significance of the allocation of funds obtained.
Champ et al. [8]	Research of the impact of an individual’s revenue limitation on the WTP values, as well as their differentiation from a perspective of knowledge/ignorance of public or environmental goods.	850	Questionnaire, the CVM.	It is necessary to realize broader research on examining the usefulness of public goods and the WTP that may create an interesting comparative database and a space for the research of other socio-economic factors that influence the WTP.
Nuva et al. [9]	Research of a rate of the WTP and the satisfaction of visitors related to conservation of resources in the national park, Gunung Gede Pangrango in Indonesia.	423	Questionnaire, interview. Logit regression model, the CVM.	Incomes, sex, and type of a settlement where visitors live represent significant factors that influence the WTP with regard to an entry fee to the national park, Gunung Gede Pangrango in Indonesia. The study recommends the increase of entry fees to the national park from a current minimum level that has not changed for 10 years. Household incomes, sex, type of housing, and price offer were significant factors that influence the amount of entry fees that visitors are willing to pay for. It is necessary to monitor the level of visitors’ satisfaction and reflect on their increasing expectations from their visits.
Zhongmin et al. [10]	Measurement of the total economic value of the ecosystem services’ recovery in the region of Ejina in China.	700	Personal interviews. The CVM, parametric model for the estimation of advantages of ecosystem recovery in Enjina.	Results show that the general public in Hei Valley would be willing to pay for the ecosystem recovery in Enjina, even this amount is lower than estimated costs for recovery.
Walpole et al. [11]	Research of a current willingness to pay rates by visitors of the national park Komodo in Indonesia.	465	Questionnaire, the CVM.	Visitors are willing to pay more than 10 times the current entry fee that represents a significant potential for income increase to recover the park.
White and Lovett [12]	Estimation of public preferences for various biotypes in the national park North York Moors in the United Kingdom by means of an environmental economic framework.	344	Questionnaire.	Authors suppose a certain level of pressure on higher fees per visit of the national parks that result from tendencies of the public authorities, which contribute less to the park’s maintenance and development. Public preferences and higher economic values will play a significant role in managing environmental resources. Future generations represent the main reason for a willingness to pay higher fees.
Reynisdottir et al. [13]	Research of the WTP rate in two natural habitats: Gullfoss waterfall and Skaftafell National Park.	252	Questionnaire. Regression analysis, the WTP.	More than 92% out of 252 respondents were willing to pay entrance fees when visiting these habitats. The results of the WTP regression analysis indicate a possible situation—a tendency to eliminate visits to such natural attraction by groups of visitors with lower incomes.
Kim et al. [14]	Research of the WTP rate during visits to natural habitats.	450	Questionnaire. The CVM and the Logit Model.	High level of willingness to pay decreases any concerns of the strategic behavior of visitors in the future. The rate of disagreement with a higher level of the WTP is subject to many discussions, especially in terms of frequent visitors as entrance fees are paid for each visit to a certain habitat.
Hadker et al. [15]	Determination of a willingness to pay when visiting the national park in Borivli.	494	Questionnaire. The CVM.	Even if India represents a developed country with a medium-low, and even low income, the visitors are willing to pay for maintaining the significant environmental habitats. Respondents responded to the WTP in various forms, both positively and negatively, while expressing their response generally and highlighting the roles of urban policies that focus on environmental protection.
Khan et al. [16]	Research of the WTP determinants and research of the influence of the improved benefits of recreation on the demand for visits to the park by visitors who are willing to pay higher fees for a better quality of environmental services.	500	Questionnaire. The CVM, the OLS Model, the Logit Model.	Significant factors are as follows: income, respondent’s age, education, and quality of recreational services. Education is considered as a significant determinant that is important in environmental protection. Consequently, the authors put pressure on the government to increase their efforts in eliminating illiteracy in the country. Quality improvement of equipment and services provided will also bring higher income to public finances, which may help future development of natural resources on a sustainable basis.
Bateman a Langford [17]	Research of the possibilities that lead to a maintenance of the Norfolk Broads, wetlands’ (broads) area, that is internationally known as being at a significant risk of saltwater flooding.	310	Questionnaire. The CVM, regression analyses.	Results show extremely practical issues that researchers need to face when estimating the value of goods—assets from its non-use. The study did not isolate this value. The WTP results in the researched areas show that only 53% of respondents agreed with higher entrance fees. Specific socio-economic indicators represent statistically significant indicators in a willingness to pay more when visiting a recreational area.
López-Mosquera [18]	Research explores gender differences in the willingness to pay for the conservation of the Monfragüe national park (Extremadura, Spain).	226	Face to face interviews. The CVM, logit model.	The results indicate that women have stronger perceived control, stronger subjective norm, and a greater willingness to pay. The study also confirmed that moral norms and perceived control are the most significant predictors of visitors’ willingness to pay intentions.
Platania and Rizzo [19]	Objective of this study was to evaluate the willingness of visitors to pay an admission ticket for the Etna Park (Italy).	110	Face to face interviews, The CVM, logit model.	Results indicate that the WTP is influenced by demographic and economic variables of respondents such as age, income, and environmental attitudes.
Baral et al. [20]	Study estimates the economic value of World Heritage Site designation for the Sagarmatha National Park, (Mount Everest) Nepal.	522	Questionnaire. The CVM, logistic regression.	Results indicate that bid amounts, gender, age, educational level, use of a guide, length of stay in the park, information about park substitutes, and knowledge about the park’s WHS designation predict visitors’ WTP.

Source: own processing based on several sources.

**Table 2 ijerph-17-03032-t002:** Number of respondents in the corrected sample.

	Price	Answer		Total
		Yes	No	
Price	3 €	85	16	101
	6 €	64	22	86
	9 €	64	30	94
	12 €	71	25	96
	15 €	71	29	100
	18 €	70	20	90
	21 €	58	34	92
	24 €	58	34	92
	27 €	47	27	74
	30 €	59	37	96
Total		647	274	921

Source: own processing based on obtained data.

**Table 3 ijerph-17-03032-t003:** The Hosmer-Lemeshow goodness of fit test.

Step	Chi-Square	Degrees of Freedom	*p*-Value
1	6.079	8	0.63

Source: own processing based on obtained data.

**Table 4 ijerph-17-03032-t004:** The logistic regression output—explanatory variable: Willingness to pay for public goods.

Variable	B	Standard Error	Wald Test	*p*-Value	Exp (B)
Autonomous constant	0.782	0.2619	8.914	0.003	2.186
Region Foreign country					
	0.309	0.1601	0.726	0.054	1.362
Bratislava	0.578	0.2000	8.357	0.004	1.783
Košice	0.387	0.1571	6.052	0.014	1.472
Prešov	0 ^a^				1
Income1701 € and more					
	0.647	0.2135	9.177	0.002	1.910
882€−1700€	0.473	0.1574	9.026	0.003	1.605
406€−882€	0.758	0.1483	26.085	0.000	2.133
Less than 405€	0 ^a^				1
Price	−0.022	0.0054	16.336	0.000	0.978

^a^ reference category; Source: own processing based on obtained data.

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
