# Peer review of "Economic Value Estimation of the Natural Heritage of the Tatra National Park"

_ijerph, 2020, doi:10.3390/ijerph17093032_

Round 1
Reviewer 1 Report
Dear Authors,
This was a practical study with detailed information. The authors used an aggregate method to obtain the economic value of the Tatra National Park. Logistic reggresion was applied to estimate the WTP model, where the data was collected from CVM questionnaires. The average WTP and expected WTP were calculated seperately. The final calculation was rather simple, as the authors used an expected WTP times historical estimated visitor numbers as the final aggregate value of the park.
The paper was in fluent writing and well arrangement. However it was difficult to judge if the approach was approprite applied.
A little bit format problems to be improved: L96, L138, L148. In L327, please check the figure legend.
L79, Reynisdottir et al., the description in final findings was a little bit different from sample size (255 or 252?), please check again.
L147, was the Price(i) in this calculation formula referred to the price that you offered to each respondent i? A little bit explanation for the conponents might be more clear.
L161, was the age structure under control? Or were all the respondents selected randomly? What was the reason to divide age groups as 19~35, 36~62, and over 62? Was it decided by clustering method?
L171, how did you divide income groups? The category boundry values were precisely, however the values seemed not very reasonable. Could you please explain a little bit for the reason?
L211, you stated an interesting concept as "strike" answers without firmly references. I suggested in this case the references were required.
L346, in the final calculation, some estimated results were cast from the calculation. However, the explanation went as "a range of values is insufficient in terms of a willingness to pay". In this case, please inform the readers about the "values" data (e.g. range, sample size, etc.), and explain your dicision reason about "insufficient" (e.g. higher than a specific value, or any other conditions).
Author Response
Dear Authors,
This was a practical study with detailed information. The authors used an aggregate method to obtain the economic value of the Tatra National Park. Logistic reggresion was applied to estimate the WTP model, where the data was collected from CVM questionnaires. The average WTP and expected WTP were calculated seperately. The final calculation was rather simple, as the authors used an expected WTP times historical estimated visitor numbers as the final aggregate value of the park.
The paper was in fluent writing and well arrangement. However, it was difficult to judge if the approach was approprite applied.
A little bit format problems to be improved: L96, L138, L148. In L327, please check the figure legend.
Checked and corrected. Figures has been redone.
L79, Reynisdottir et al., the description in final findings was a little bit different from sample size (255 or 252?), please check again.
Checked and corrected – correct sample size is 252.
L147, was the Price(i) in this calculation formula referred to the price that you offered to each respondent i? A little bit explanation for the conponents might be more clear.
Yes, Price(i) in this calculation formula referred to the price we offered to each respondent i. This information was added to text.
L161, was the age structure under control? Or were all the respondents selected randomly? What was the reason to divide age groups as 19~35, 36~62, and over 62? Was it decided by clustering method?
All respondents were selected randomly. Primary we had more age groups in questionnaire, options were: â–¡ 15-18, â–¡ 19-24, â–¡ 25-30, â–¡ 31-35, â–¡ 36-40, â–¡ 41-45, â–¡ 46-50, â–¡ 55-62, â–¡ 64+, but with aim to better fitting the model, we reduced the age clusters on three. Clustering was performed by optimal binnig method with respect to WTP.
L171, how did you divide income groups? The category boundry values were precisely, however the values seemed not very reasonable. Could you please explain a little bit for the reason?
Following was added to text: 882 € represented average nominal wage in Slovakia in 2016. Another group of respondents ranged from 883 € to 1700 € (31 % of respondents). As mentioned above, 883 € represented average nominal wage in 2016 and 1700 was a median wage of employees within industries with highest wages in Slovakia in 2016 (IT sector, finance and insurance, power engineering).
L211, you stated an interesting concept as "strike" answers without firmly references. I suggested in this case the references were required.
Reference has been added.
L346, in the final calculation, some estimated results were cast from the calculation. However, the explanation went as "a range of values is insufficient in terms of a willingness to pay". In this case, please inform the readers about the "values" data (e.g. range, sample size, etc.), and explain your dicision reason about "insufficient" (e.g. higher than a specific value, or any other conditions).
We meant that the range of options for WTP were from 0 – 30 Euros. Maybe we should make the interval of possible prices wider. We explained the given opinion already in conclusion, limitations of the study: The submitted study has two limitations. Firstly, the scale of hypothetical entrance fees to TANAP could have been wider.
Additionally, upon request of the reviewer, we added following into L350: This result may not be considered as aggregate value of TANAP, because a range of values is insufficient in terms of a willingness to pay (scale of hypothetical entrance fees to TANAP could have been wider in the questionnaire).
We hope that we answered all questions of reviewer.
We would like to thank review for his/her suggestions that added value to our paper.
Reviewer 2 Report
The research followed well the single bounded dichotomous contingent valuation method. The calculation process is demonstrated clearly. And the study site is of unique and significance, in terms of historical tourism site. However, the research keep many old and had well-known common knowledge related to the contingent valuation. There are no much creation. The back ground and the significance and the needs for this research is not sufficiently and clearly shown in this paper. The same research method and calculating process can be duplicated in any place. The author do not provided strong and enough about the need to do this research. There are no clear presentation of research design.
For improving the present paper to show a strong value, there are several suggestions for the authors’ consideration and for their further improvement.
- The section of material and methods are well presented. The research is based on a sound and wide accepted method.
- The structure of the paper need to be modified to present the background, the methodology, the research design, the commit of the survey, the research results, and then discussion and conclusion.
- What to be evaluated? The research needs to provide more significant attribute of the study site, in order to show the needs to be protected by UNESCO, etc. What eco-service the site provided? What is the threats? What kind of ecological services to be protected? … Then, we can connect the “target” goods for this valuation, and connected the target goods with the important eco-services it provided.
- The design of this research is still needs to be clearly provided in a single section, even though it is scattered in many different places. Line 242-265 should be appeared in the missing section of “research design”.
- What is the design and the process committed the survey? The authors are suggested to neatly and clearly provide the detailed process of survey. It seems missing details about how the research survey was conducted.
- Line 40 is not very clearly express the meaning of the authors. The three branches of the research is much more related to “the valuation for non-marketed goods”, rather than recognition of that goods. The three branches is commonly introduced in related textbook. So it seems not of value to be introduced here. If it need to be appear here, the authors should verify the
- In order to make sense in Line 63-64, a reversion is suggested. It is a qualification “by using CVM evaluation technique”.
- The literature is good to list related papers with table. But it is not enough for a literature to be appeared in a journal paper. The authors are suggested to have comparison (with current research) and comments, and let the readers recognize how this research is based on and connected to the literature. After all, the authors are suggested to increase comments on the literature.
- How do the authors create the asking values appear in Line 157. Please clearly describe. In Lin 165-175, the description for the sample on the demographic structure is not clear.
- In Table 4 the “parameter” should be changed to be the “variable”; and in fact, “B” in this table is the vector of parameters.
The reviewer hope you can soundly and successfully provide the value of the significant natural assets and make them really protected.
Author Response
For improving the present paper to show a strong value, there are several suggestions for the authors’ consideration and for their further improvement.
The section of material and methods are well presented. The research is based on a sound and wide accepted method.
The structure of the paper need to be modified to present the background, the methodology, the research design, the commit of the survey, the research results, and then discussion and conclusion.
Sections Background of the study and Research design has been added.
What to be evaluated? The research needs to provide more significant attribute of the study site, in order to show the needs to be protected by UNESCO, etc. What eco-service the site provided? What is the threats? What kind of ecological services to be protected? … Then, we can connect the “target” goods for this valuation, and connected the target goods with the important eco-services it provided.
Added section Background of the study - we did our best to present Tatra national park and its importance for biodiversity.
The design of this research is still needs to be clearly provided in a single section, even though it is scattered in many different places. Line 242-265 should be appeared in the missing section of “research design”.
We added section Research design and described the design of the research.
What is the design and the process committed the survey? The authors are suggested to neatly and clearly provide the detailed process of survey. It seems missing details about how the research survey was conducted.
We added section Research design and described the design of the research.
Line 40 is not very clearly express the meaning of the authors. The three branches of the research is much more related to “the valuation for non-marketed goods”, rather than recognition of that goods. The three branches is commonly introduced in related textbook. So it seems not of value to be introduced here. If it need to be appear here, the authors should verify the – recognition replaced by valuation
In order to make sense in Line 63-64, a reversion is suggested. It is a qualification “by using CVM evaluation technique”
Reformulated as proposed by reviewer
The literature is good to list related papers with table. But it is not enough for a literature to be appeared in a journal paper. The authors are suggested to have comparison (with current research) and comments, and let the readers recognize how this research is based on and connected to the literature. After all, the authors are suggested to increase comments on the literature.
We expanded the literature review part with relevant literature (WTP in context of natural parks). We also increased comments on the literature.
How do the authors create the asking values appear in Line 157. Please clearly describe. Question was: Imagine that the fee should be collected for entrance to the Tatra National Park. Would you be willing to pay an annual amount (here the fee varied from 3 – 30 EUROS) for entering this area, in exchange for a guarantee that no interference with the integrity of TANAP (further construction of hotels, cable cars or ski resorts) will take place and your payment will also cover repairs and maintenance of the Tatra National Park?
We added this to the text, 179.
In Lin 165-175, the description for the sample on the demographic structure is not clear.
We reformulated sentences to make it clearer.
In Table 4 the “parameter” should be changed to be the “variable”; and in fact, “B” in this table is the vector of parameters.
We changer expression Parameter to the Variable
The reviewer hope you can soundly and successfully provide the value of the significant natural assets and make them really protected.
Thank a lot to reviewer for his/her constructive and professional suggestions that made paper more valuable. We integrated as much ideas of reviewer as possible.
Round 2
Reviewer 1 Report
Dear Authors,
Thank you for the modifications and your kindly responses to my confusion on some descriptions, and the additional information was very helpful. The Background section helped readers from other study field to understand your research, thank you.
A little bit format problems to be improved: L211, L222, L244. From L347 to L351, I understand you were trying to fit the table 4 in this section, a little bit arrangement for the table and description might be better? Please check on these lines and help readers focus on your paper.
Best regards.
Author Response
We are thankful to reviewer for additional comments. We checked the format problems and corrected them. We also arranged Table 4 and rearranged text that concerns in.
Reviewer 2 Report
The revise version is improved greatly. There still some minor spell errors as indicated in what followed.
- Line 326: 35.2 should be 5.2
- L 105: There are two "by". However, the first "by" should be removed.
- In Abstract, the value is intrinsic existing there, in my opinions. It is "estimated" or is "revealed" through the present research, rather than "is determined". Please consider to revise.
- Line 19: the meaning is quite vogue as the authors wrote "every additional price increase". According to the research design, it is increase 3 Euros for every incremental additional bidding.
Author Response
We are thankful to reviewer for additional comments.
- Line 326: 35.2 should be 5.2
Checked and corrected.
- L 105: There are two "by". However, the first "by" should be removed.
The correct form is: The answer might be outlined by presented study … not as it was written before: The answer might by outlined by presented study
- In Abstract, the value is intrinsic existing there, in my opinions. It is "estimated" or is "revealed" through the present research, rather than "is determined". Please consider to revise.
Determine changed for Estimate.
- Line 19: the meaning is quite vogue as the authors wrote "every additional price increase". According to the research design, it is increase 3 Euros for every incremental additional bidding.
We reformulated the sentence as following: For every increase of fee by 3€, the willingness to pay for entry to the Tatra National Park decrease by 2.2%.